# Neonatal near miss in a State of Northeastern Brazil: Near miss characterization and determinants

**Daiane Porto Nery**[1], **Amanda Cristina de Souza Andrade**[2], **Daniela Silva Rocha**[1], **Vanessa Moraes Bezerra**[1]*

**1** Federal University of Bahia, Multidisciplinary Institute of Health, Vitória da Conquista, Bahia, Brazil, **2** René Rachou Institute, Oswaldo Cruz Foundation (FIOCRUZ-Minas), Belo Horizonte, Minas Gerais, Brazil

* vanessaenut@gmail.com

## Abstract

This study aimed to evaluate neonatal near miss (NNM) in a state in northeastern Brazil and to identify possible sociodemographic, obstetric, childbirth, and healthcare-related determinants. This cohort study of live births was conducted from 2012 to 2020 in Bahia, Brazil (n = 1,821,384), using data retrieved from national health information systems (Live Birth Information System and Mortality Information System). The criteria for defining NNM were gestational age < 32 weeks, birth weight <1,500 grams, 5-minute Apgar score <7, and congenital malformations. Logistic regression models with hierarchical entry of variables were adjusted, and the odds ratio was estimated with 95% confidence intervals. The NNM rate was 30.4/1,000 LB. The following risk factors for NNM were identified: newborn sex (male); maternal age (10–19 years and 35 years or older); low maternal schooling; not having a partner; having one or more fetal losses/miscarriages; multiple pregnancies; having had fewer than six prenatal care appointments; absence of labor induction; having had a cesarean section and non-cephalic fetal presentation. The protective factors for NNM were being primiparous and having had one or more previous vaginal or cesarean deliveries. The NNM rate was associated with sociodemographic, obstetric, childbirth, and healthcare factors. Addressing this complex event requires include everything from breaking down social disparities to a robust health system that responds to calls for treatment and rehabilitation and consistently working on promotion and prevention actions in maternal and child health.

**Data availability statement:** The data is not in the public domain. It was provided by the Secretary of State of Bahia. Therefore, the data will be made available upon reasonable request. Contact the state health department: divep. coap@saude.ba.gov.br.

**Funding:** The author(s) received no specific funding for this work.

**Competing interests:** The authors have declared that no competing interests exist.

## Introduction

In recent years, infant and neonatal mortality has been at the center of health debates due to its adverse impacts. This indicator is sensitive to populations living and health conditions and can reveal inequalities, especially in territories where its frequency is high [1,2]. Although neonatal mortality has significantly dropped in Brazil, the rate distribution varies across regions, pointing to inequalities [2,3].

The North and Northeast have had the worst neonatal mortality rates in recent years, revealing perpetuating inequalities. In 2018, while the early and late neonatal mortality rates were 5.3/1,000 live births and 1.9/1,000 live births in the country's South, the Northeast recorded 8.3 and 2.4 per 1,000 live births [2,3]. The Northeast is one of the poorest regions in the country. Situations of racial and economic segregation are identified, and they can influence their population's living and health conditions. In this region, the state of Bahia stands out with the most significant number of people who self-identify as Black or brown (together, they make up the Black population) [4]. Additionally, the Northeast has one of the largest concentrations of people who benefit from income transfer programs. A significant portion of these beneficiaries reside in Bahia [5].

From the perspective of the vulnerabilities of mothers and their newborns, it is also necessary to consider those newborns who experienced severe conditions that could have resulted in death but survived. This situation is labeled as neonatal near miss (NNM), a term used in the literature to identify and elucidate cases of newborns who almost died but survived the neonatal period either by chance or due to the high-quality care they received [6,7]. Once exposed to NNM, newborns may struggle in their growth and development throughout their lives and endure comorbidities and disabilities [7]. Early recognition of these individuals through this indicator is crucial for population health planning [6].

The NNM rates identified from a systematic review of studies from several countries ranged from 11.0 to 72.5 per 1,000 live births. These variations are related to the varying criteria used and the different contexts evaluated [7]. Pragmatic, clinical, and management criteria are among the most widely used to classify NNM [6,7]. Due to data availability, adopting pragmatic criteria, such as gestational age, birth weight, 5-minute Apgar score of life, and congenital malformations [6] is recommended.

A Brazilian study employing pragmatic criteria identified varying NNM rates in different regions of the country: 29.7/1,000 live births in a municipality in the Southeast region, 38.8/1,000 live births in a capital in the Northeast (São Luis do Maranhão), and 30.3/1,000 live births in the city of Pelotas, located in the South region of the country [8]. Variations in the NNM social determinants were also observed [8]. Maternal, gestational, and prenatal care sociodemographic characteristics are the variables investigated as potential NNM determinants [8,9]. The need to study this territory is justified given the the well-documented social inequalities in the Northeast region and the recommended NNM monitoring for health planning [7]. The present study aims to evaluate the association between the neonatal near miss (NNM) rate and sociodemographic, obstetric, childbirth, and healthcare factors in a state in Northeastern Brazil.

## Methods

### Study setting

This was a cohort study of live births from Bahia, Brazil, from 2012 to 2020. The data were collected on November 28, 2022 from the Live Birth Information System and the Mortality Information System provided by the State Health Secretariat for use in this study.

### Ethics statement

The Research Ethics Committee of the Multidisciplinary Institute of Health of the Federal University of Bahia (IMS/UFBA) approved this research under Opinion N°5,623,213. All ethical principles for research involving human subjects were met. As this was a study based on secondary data, the ethics committee waived the need for consent from participants.

### Data

The data from both information systems were subjected to a pre-processing phase to adjust and standardize "variables contents". Duplicate records were identified based on the number of the Live Birth Certificate or Death Certificate and were subsequently excluded.

We performed deterministic linkage of the databases to identify deaths up to the 27th day of life after birth using the Live Birth Certificate number as a common field for both systems. For records whose Live Birth Certificate number was missing, we performed probabilistic linkage using the mother's name, sex, and date of birth of the newborn as variables. We used information from the Mortality Information System to identify neonatal deaths, which, once identified, were excluded from the database. All live newborns who survived until the 27th day of life after birth were included in the study (n = 1,821,384).

### Variables

The outcome variable was NNM, dichotomized as "yes" and "no". NNM cases were defined based on at least one of the criteria adapted from Silva et al. (2017) [10], namely: birth weight <1,500 g, gestational age (GA) <32 weeks, Apgar <7 at the 5-minute Apgar score, and congenital malformation. We calculated NNM rates, defined as the number of NNM cases divided by the total number of live births multiplied by one thousand.

The independent variables were the sociodemographic factors: newborn's sex; mother's age group (10–19 years, 20–34 years, and 35 years or more); mother's maternal race/ethnicity (white, brown/black, and yellow/indigenous); schooling years (0–7 years, 8–11 years, and 12 years or more); maternal employment status (employed/not employed) and marital status (with or without a partner). The obstetric and childbirth factors were the number of previous pregnancies, the number of previous normal deliveries, previous cesarean sections, the number of live and stillbirths, and pregnancy type (single or multiple). The healthcare factors were the number of prenatal care appointments (<6 or ≥6), induced labor (yes or no), delivery type (cesarean or vaginal), and fetal presentation (cephalic and non-cephalic).

### Statistical analysis

We performed univariate analysis with an estimated OR and calculated the respective 95% confidence intervals (95%CI) to verify the factors associated with NNM. We employed logistic regression, and variables with a p-value <0.20 were selected for entry in the multiple model. Before insertion into the multiple model, the multicollinearity of the independent variables was assessed using the variance inflation factor (VIF). Variables with a VIF value ≥ 10 [11] were considered collinear, which was observed for the variable number of living children (VIF = 18.33).

We adopted the hierarchical entry of variables in blocks (Kale et al., (2017), Modes et al., (2024) [12,13], in the following order: Block 1 (Sociodemographic factors), Block 2 (Obstetric and childbirth factors), and Block 3 (Healthcare). The

variables of the most distal blocks remained as adjustment factors for the hierarchically lower blocks (Fig 1). We compared the models through the Akaike criterion (AIC), adopting a significance level of 5%. All analyses were performed in the Stata program version 15.1 (Stata Corp., College Station, United States).

## Results

Among the 1,821,384 live births who survived the neonatal period, 55,285 met at least one NNM criterion, 55,285 presented at least one NNM criterion and the NNM rate was 30.4/1,000 live births. Among the NNM criteria, 40.7% (n = 21,442) a gestational age of less than 32 weeks, 32.2% (n = 16,949), had 5-minute Apgar score of less than 7, 29.4% (n = 16,259) had a birth weight <1,500 g, and 21.5% (n = 11,544) had congenital malformation. The mean (standard deviation) of 5-minute Apgar score was 7.6 (0.01) and the mean birth weight was 2,397.4 g (4.53).

Regarding sociodemographic characteristics, most mothers had male children had male children, were 20–34 years old, had 8–11 years of schooling, were black or brown, did not work, and had a partner. A higher likelihood of NNM was observed among mothers with male children, mothers at age extremes, those with lower schooling levels, those who did not work, and those who did not have a partner (Table 1).

In terms of obstetric characteristics, most mothers had had one or more previous pregnancies, had no previous cesarean or vaginal deliveries, had one or more live births, had no stillbirths, and had singleton pregnancies. Regarding

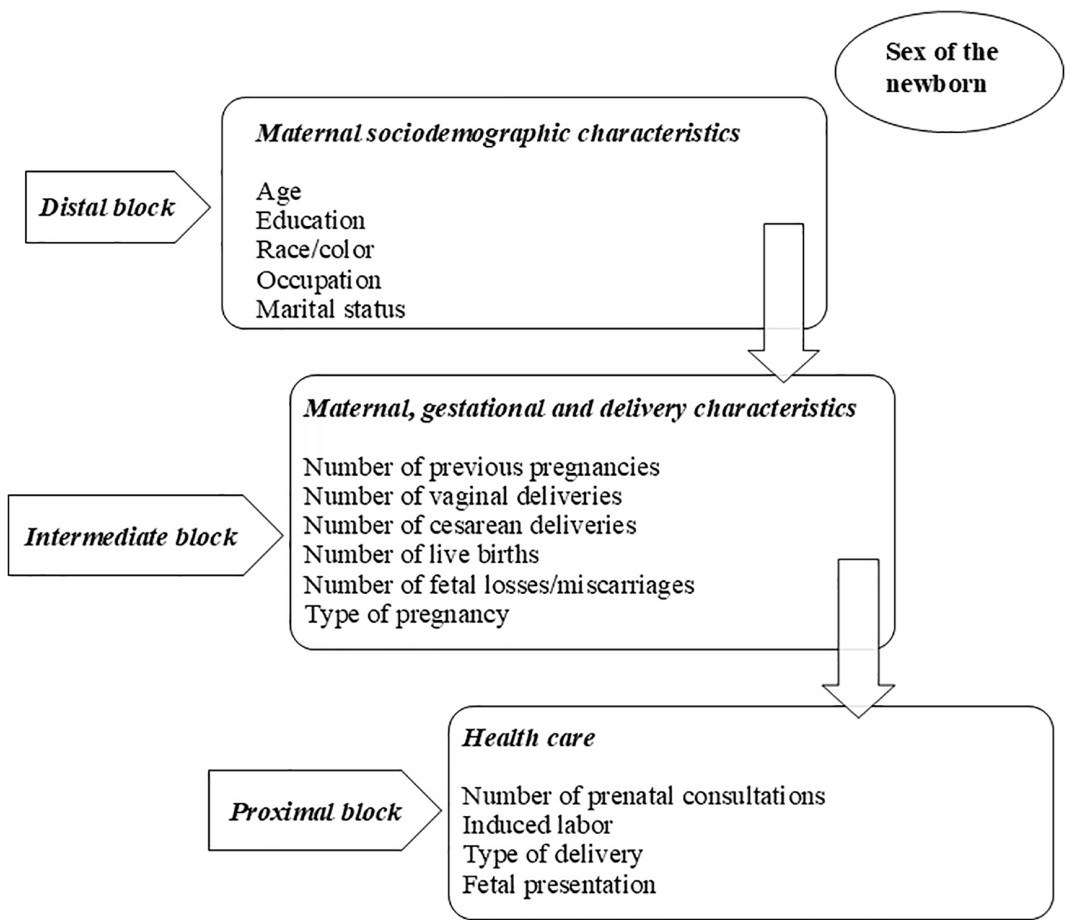

**Fig 1. Conceptual Model for neonatal near miss adapted from Kale et al. (2017) [12] and Modes et al. (2024) [13].**

**Table 1. Distribution of the studied population, neonatal near miss rates (per 1,000 live births) and crude analysis (Odds Ratio) of the association between sociodemographic variables and NMN. Bahia, Brazil, 2012-2020.**

| | Live births | | Neonatal near miss | |
|---|---|---|---|---|
| Variables | n | % | NMN rates | OR (CI95%) |
| **Sociodemographic factors** | | | | |
| **Newborn sex** | | | | |
| Female | 890,489 | 48.9 | 28.8 | 1.00 |
| Male | 930,670 | 51.1 | 31.7 | 1.10 (1.08-1.12) |
| **Maternal age** | | | | |
| 20 to 34 years | 1,229,928 | 67.53 | 28.0 | 1.00 |
| 10 to 19 years | 350,981 | 19.27 | 35.4 | 1.27 (1.25-1.30) |
| 35 years or older | 240,460 | 13.20 | 35.0 | 1.26 (1.23-1.29) |
| **Maternal education** | | | | |
| 12 years or more | 226,875 | 13.03 | 27.8 | 1.00 |
| 8 to 11 years | 1,016,489 | 58.37 | 29.6 | 1.07 (1.04-1.10) |
| 0 to 7 years | 498,202 | 28.61 | 33.3 | 1.21 (1.17-1.24) |
| **Work** | | | | |
| Yes | 684,396 | 37.58 | 30.0 | 1.00 |
| No | 1,136,988 | 62.42 | 30.6 | 1.02 (1.01-1.04) |
| **Marital status** | | | | |
| With partner | 912,475 | 51.6 | 29.0 | 1.00 |
| Without partner | 855,840 | 48.4 | 31.9 | 1.10 (1.08-1.12) |

\* NMN rate: Neonatal near miss rate; 95%CI: 95% confidence interval; OR: Odds Ratio.

healthcare, most mothers had six or more prenatal care appointments, had not had induced labor, had vaginal deliveries, and had a non-cephalic fetal presentation. Women who had at least one previous pregnancy, a previous history of vaginal or cesarean deliveries, and a history of live births were less likely to have NNM. Positive associations with NNM were found in women with a history of one or more fetal losses/miscarriages, those with multiple pregnancies, fewer than six prenatal care appointments, non-induced labor, cesarean delivery, and non-cephalic fetal presentation (Table 2).

In the multivariable analysis, the variables positively associated with NNM in Model 1 were: male newborn (OR=1.10; 95% CI = 1.08-1.12); maternal age (10–19 years: OR=1.22; 95%CI = 1.19-1.25 or 35 years or older: OR=1.26; 95%CI = 1.23-1.30); low schooling level (8–11 years: OR=1.06; 95%CI = 1.03-1.09 and 0–7 years: OR=1.15; 95%CI = 1.12-1.20); those without a partner (OR=1.07; 95%CI = 1.05-1.09). In Model 2, the following were negatively associated with NNM: having had a previous pregnancy (OR=0.90; 95%CI = 0.86-0.94) and previous history of vaginal deliveries (one delivery: OR=0.92; 95%CI = 0.88-0.96 and 2 or more: OR=0.85; 95%CI:0.80-0.91) or cesarean sections (one delivery: OR=0.86; 95%CI = 0.83-0.90 and 2 or more: OR=0.87; 95%CI:0.81-0.94); and positively associated with a history of one (OR=1.13; 95%CI = 1.09-1.17) or two or more (OR=1.35; 95%CI = 1.27-1.43) fetal losses/miscarriages and multiple pregnancies (OR=4.09; 95%CI = 3.93-4.26). In Model 3, positive associations were observed with less than six prenatal care appointments (OR=2.33; 95%CI = 2.28-2.38); non-induced labor (OR=1.18; 95%CI = 1.14-1.21); cesarean delivery (OR=1.05; 95%CI = 1.02-1.07) and non-cephalic fetal presentation (OR=2.39; 95%CI = 2.30-2.49) with the event studied (Table 3).

## Discussion

An NNM rate of 30.4/1,000 live births was identified among live births in Bahia from 2012 to 2020. Regarding the factors investigated, the sex of the newborn (male), mothers aged 10–19 years or 35 years or older, low schooling level, and

**Table 2. Distribution of the study population, neonatal near miss rates (per 1,000 live births) and crude analysis (Odds Ratio) of the association between obstetric and delivery variables with NMN. Bahia, Brazil, 2012-2020.**

| | Live births | | Neonatal near miss | |
|---|---|---|---|---|
| **Variables** | **n** | **%** | **NMN rates** | **OR (CI95%)** |
| **Obstetric and childbirth factors** | | | | |
| **Number of previous pregnancies** | | | | |
| 0 | 515,245 | 33.12 | 33.4 | 1.00 |
| 1 | 505,796 | 32.51 | 27.7 | 0.82 (0.81-0.84) |
| 2 | 273,682 | 17.59 | 29.2 | 0.87 (0.84-0.89) |
| 3 or more | 261,043 | 16.78 | 33.3 | 0.99 (0.97-1.02) |
| **Number of vaginal births** | | | | |
| 0 | 770,706 | 51.91 | 31.9 | 1.00 |
| 1 | 375,735 | 25.31 | 28.7 | 0.90 (0.88-0.92) |
| 2 or more | 338,149 | 22.78 | 30.9 | 0.97 (0.94-0.99) |
| **Number of cesarean deliveries** | | | | |
| 0 | 1,114,536 | 77.88 | 31.9 | 1.00 |
| 1 | 250,862 | 17.53 | 26.8 | 0.84 (0.81-0.86) |
| 2 or more | 65,785 | 4.6 | 30.0 | 0.94 (0.90-0.98) |
| **Number of children born alive** | | | | |
| 0 | 585,756 | 37.86 | 34.0 | 1.00 |
| 1 | 538,762 | 34.82 | 26.9 | 0.78 (0.77-0.80) |
| 2 or more | 422,793 | 27.32 | 30.4 | 0.89 (0.87-0.91) |
| **Number of fetal losses/miscarriages** | | | | |
| 0 | 1,111,029 | 76.96 | 30.2 | 1.00 |
| 1 | 265,842 | 18.41 | 32.3 | 1.07 (1.05-1.10) |
| 2 or more | 66,750 | 4.62 | 39.8 | 1.33 (1.28-1.38) |
| **Type of pregnancy** | | | | |
| Single | 1,781,013 | 98.04 | 28.8 | 1.00 |
| Twin or more | 35,672 | 1.96 | 107.4 | 4.06 (3.92-4.20) |
| **Health care** | | | | |
| **Number of prenatal consultations** | | | | |
| 6 or more | 1,277,943 | 74.34 | 23.0 | 1,00 |
| Less than 6 | 441,109 | 25.66 | 50.6 | 2.26 (2.22-2.30) |
| **Induced labor** | | | | |
| Yes | 310,807 | 18.48 | 26.1 | 1.00 |
| No | 1,370,844 | 81.52 | 32.0 | 1.23 (1.20-1.26) |
| **Type of birth** | | | | |
| Vaginal | 1,016,166 | 55.92 | 29.7 | 1.00 |
| Cesarean section | 800,968 | 44.08 | 31.2 | 1.05 (1.04-1.07) |
| **Fetal presentation** | | | | |
| Cephalic | 1,707,569 | 96.22 | 28.4 | 1.00 |
| Non-cephalic | 67,128 | 3.78 | 81.4 | 3.03 (2.94-3.12) |

* NMN rate: Neonatal near miss rate; 95%CI: 95% confidence interval; OR: Odds Ratio.

**Table 3. Multiple logistic regression analysis of factors associated with neonatal near miss. Bahia, Brazil, 2012-2020.**

| Variables | Model 1 OR | CI95% | Model 2 OR | CI95% | Model 3 OR | CI95% |
|---|---|---|---|---|---|---|
| **Block 1: Sociodemographic factors** | | | | | | |
| **Newborn sex** | | | | | | |
| Female | 1.00 | | 1.00 | | 1.00 | |
| Male | 1.10 | 1.08-1.12 | 1.10 | 1.08-1.13 | 1.11 | 1.09-1.14 |
| **Maternal age** | | | | | | |
| 20 to 34 years | 1.00 | | 1.00 | | 1.00 | |
| 10 to 19 years | 1.22 | 1.19-1.25 | 1.18 | 1.15-1.21 | 1.07 | 1.04-1.10 |
| 35 years or older | 1.26 | 1.23-1.30 | 1.23 | 1.19-1.27 | 1.25 | 1.21-1.29 |
| **Maternal education** | | | | | | |
| 12 years or more | 1.00 | | 1.00 | | 1.00 | |
| 8 to 11 years | 1.06 | 1.03-1.09 | 1.11 | 1.07-1.15 | 1.09 | 1.05-1.13 |
| 0 to 7 years | 1.15 | 1.12-1.20 | 1.26 | 1.21-1.31 | 1.15 | 1.10-1.20 |
| **Marital status** | | | | | | |
| With partner | 1.00 | | 1.00 | | 1.00 | |
| Without partner | 1.07 | 1.05-1.09 | 1.06 | 1.04-1.08 | 0.98 | 0.96-1.01 |
| **Block 2: Obstetric and childbirth factors** | | | | | | |
| **Number of previous pregnancies** | | | | | | |
| 0 | | | 1.00 | | 1.00 | |
| 1 | | | 0.90 | 0.86-0.94 | 0.86 | 0.82-0.90 |
| 2 | | | 0.95 | 0.89-1.02 | 0.89 | 0.82-0.95 |
| 3 or more | | | 1.01 | 0.93-1.10 | 0.86 | 0.79-0.95 |
| **Number of vaginal births** | | | | | | |
| 0 | | | 1.00 | | 1.00 | |
| 1 | | | 0.92 | 0.88-0.96 | 0.90 | 0.86-0.94 |
| 2 or more | | | 0.85 | 0.80-0.91 | 0.82 | 0.76-0.88 |
| **Number of cesarean deliveries** | | | | | | |
| 0 | | | 1.00 | | 1.00 | |
| 1 | | | 0.86 | 0.83-0.90 | 0.83 | 0.80-0.87 |
| 2 or more | | | 0.87 | 0.81-0.94 | 0.83 | 0.76-0.90 |
| **Number of fetal losses/miscarriages** | | | | | | |
| 0 | | | 1.00 | | 1.00 | |
| 1 | | | 1.13 | 1.09-1.17 | 1.19 | 1.15-1.24 |
| 2 or more | | | 1.35 | 1.27-1.43 | 1.44 | 1.36-1.53 |
| **Type of pregnancy** | | | | | | |
| Single | | | 1.00 | | 1.00 | |
| Twin or more | | | 4.09 | 3.93-4.26 | 3.14 | 3.00-3.29 |
| **Block 3: Health care** | | | | | | |
| **Number of prenatal consultations** | | | | | | |
| 6 or more | | | | | 1.00 | |
| Less than 6 | | | | | 2.33 | 2.28-2.38 |
| **Induced labor** | | | | | | |
| Yes | | | | | 1.00 | |
| No | | | | | 1.18 | 1.14-1.21 |
| **Type of birth** | | | | | | |
| Vaginal | | | | | 1.00 | |

*(Continued)*

**Table 3.** (Continued)

| Variables | Model 1 | | Model 2 | | Model 3 | |
|---|---|---|---|---|---|---|
| | OR | CI95% | OR | CI95% | OR | CI95% |
| Cesarean section | | | | | 1.05 | 1.02-1.07 |
| **Fetal presentation** | | | | | | |
| Cephalic | | | | | 1.00 | |
| Non-cephalic | | | | | 2.39 | 2.30-2.49 |
| **Akaike criterion** | | | | | | |
| | 464163.70 | | 353827.30 | | 315921.1 | |

*Model 1: adjusted between sociodemographic variables; Model 2: adjusted between sociodemographic and obstetric and childbirth variables; Model 3: adjusted between sociodemographic, obstetric and childbirth and health care variables; OR: Odds Ratio; 95%CI: 95% confidence interval.

those without a partner were positively associated with NNM. Among the obstetric characteristics, having had a previous pregnancy and one or more vaginal or cesarean deliveries were negatively associated with NNM. In contrast, one or more fetal losses/miscarriages and multiple pregnancies were positively associated with NNM. Regarding healthcare, less than six prenatal care appointments, non-induced labor, cesarean section, and non-cephalic fetal presentation were positively associated with NNM.

Variations in NNM rates were observed when compared with other regions of the country that used pragmatic criteria similar to those adopted in the present study [8–10]. A rate of 22.8 was observed in Cuiabá, Mato Grosso [9], 29.7 in Ribeirão Preto, São Paulo, 38.8 in São Luiz, Maranhão, and 30.3 in Pelotas, Rio Grande do Sul [8], all rates multiplied by 1,000 live births. A study conducted in a developed country found that 17.2/1,000 live births of the newborns investigated were classified as having NNM [14]. Differences in the occurrence of NNM between countries and Brazilian regions may reveal social inequalities. Comparing the rates with other studies is challenging because of the varying criteria employed to classify NNM [6,7].

Since NNM is an essential indicator for situational diagnosis, planning and management in maternal and child health [6,7], understanding its social determinants can enhance the effectiveness of its use. Regarding the determinants studied, male gender was significantly associated with NNM in the present study, corroborating a study conducted in Ethiopia [15]. There is evidence that boys have a higher mortality rate and a lower life expectancy than their counterparts. There is little information about the relationship between gender and adverse events in childhood. The literature suggests that males are more genetically vulnerable, which, for example, increases the risk of developing hemorrhages and miscarriages [16].

Maternal age (mothers aged 10–19 or over 35 or older), lower schooling levels, and those without a partner were more likely to have NNM. The association between maternal age extremes and NNM has been observed in the literature [8,9,14]. Several factors, such as physiological and psychological immaturity, less knowledge about sexual and reproductive issues [17], and late and insufficient initiation of prenatal care follow-up [18], can explain adverse outcomes among adolescent mothers. Furthermore, teenage pregnancy occurs more frequently in populations that are victims of social inequalities [19], and the region of the present study stands out as a Brazilian territory with more significant vulnerability when compared to other regions of the country [4].

Regarding older ages, the observed association can be understood from the perspective of overlapping diseases acquired throughout life. Having some previous pathological condition was positively associated with severe neonatal outcomes in a study conducted in Africa [20], which may contribute to more significant exposure of the mother and baby to several complications, such as premature birth, low birth weight, and a higher risk of death in the neonatal period [9]. Furthermore, the distribution of chronic noncommunicable diseases (NCDs) is heterogeneous among women with different socioeconomic conditions. Women who benefited from an income transfer program in Brazil (more present in the North and Northeast regions) had a higher prevalence of risk factors for NCDs [21]. In the same direction, the more significant

presence of NCDs in lower schooling levels stands out. As with neonatal mortality, low maternal education has emerged as a risk factor for adverse outcomes in the neonatal period, often due to barriers to access to healthcare [8,9,22].

Mothers without a partner were also more likely to have children classified as having NNM. The literature indicates benefits in paternal involvement during pregnancy, childbirth, and postpartum, such as greater adherence to prenatal care, support in decision-making, and caring for the baby [23]. On the other hand, mothers without a partner are more susceptible to stress, anxiety, and depression during pregnancy. Additionally, the frequency of exposure to other risk factors increases, such as illicit drug use and inadequate prenatal care, which contributes to adverse outcomes [8,9]. We should underscore the little or no influence of sociodemographic factors explored on NNM [10] in more privileged Brazilian regions. This finding reveals social inequalities that adversely impact the living conditions and health of the women and newborns investigated.

Regarding pregnancy history, women with a previous pregnancy and who had one or more births (vaginal or cesarean) were less likely to have children classified as NNM. Previous studies found different associations between the number of previous pregnancies, in which having three or more previous pregnancies increased the likelihood of having NNM [13,24]. No association was observed between the number of previous births and NNM [9,23]. The explanation for the findings of the present study is seemingly related to maternal age. In the present study, it was observed that among adolescent mothers, 73.2% were nulliparous. Considering the risks related to teenage pregnancy [17–19], having had a previous pregnancy and a history of previous births seems to be closely linked to a more favorable maternal age for pregnancy.

Pregnant women with a previous history of one or more fetal losses/miscarriages were more likely to have a child classified as NNM in the present study. Advanced maternal age, previous history of premature birth, cesarean section, or gestational diabetes [25] stand out as a risk factor for spontaneous abortion. In this sense, discussing gestational risk, which includes an obstetric history of abortion, needs to be at the heart of this issue, transcending clinical criteria and reaching social inequalities in health. Risk factors often overlap and increase each other. Lower schooling levels, pregnant women as family heads, and being a beneficiary of social benefits were positively associated with high gestational risk [25,26].

Regarding the type of pregnancy, newborns from multiple pregnancies were more likely to experiencing NNM. This finding corroborates other studies [13,15]. Twin pregnancies commonly progress to prematurity and low birth weight. Also, a higher risk of morbimortality is observed among newborns from multiple pregnancies when compared to singleton pregnancies [27]. This event requires a structured Health Care Network (RAS). Primary Health Care organizes and coordinates the RAS and faces some weaknesses. A study conducted in the Northeast identified a fragmented regional network with difficult access to specialized care [28], which may impact the healthcare of its maternal and child population.

The present study identified a higher NNM rate among mothers with fewer than six prenatal care appointments, and a similar result was observed in other regions of the country [9,13]. Although there have been advances in prenatal care coverage in Brazil [22], we note persistent regional disparities, such as fewer opportunities for access to health services in the Brazilian North and Northeast [29]. The quality and adequacy of prenatal care is a factor that should also be considered. Women living in the poorest Brazilian regions tend to receive lower-quality prenatal care [30].

The positive association between non-induced labor and NNM observed in the present study has been poorly explored in the literature [9,13]. Lower exposure to interventions such as non-induced labor should mean ensuring better care for the mother-baby binomial. However, this finding may be related to the reduction in care offered to the pregnant woman, who may be under-assisted during labor regarding necessary interventions, producing adverse outcomes [31].

Regarding delivery characteristics, newborns by cesarean section were more likely to suffer NNM. This result has already been established in the literature [8,10,14]. The number of cesarean sections is high in Brazil. The literature indicates that poorly indicated cesarean sections can adversely affect the mother/child binomial, such as prematurity and maternal morbidity [32]. In order to eliminate the indiscriminate indication of cesarean sections, we advocate for the promotion of continuous support by doulas, partners, and health professionals to women during labor and delivery [32].

The likelihood of NNM was also higher among newborns with a non-cephalic fetal presentation. These data are still little explored in the literature [9,13,24]. Conducting delivery in circumstances where the baby's positioning is less favorable

adds complexity to the event. The best route of delivery to assist in these situations is still controversial. It is argued that normal delivery is safe under these conditions when the obstetrician attending the birth is experienced and meets strict selection criteria for its indication. It is also recommended that normal delivery be preferred due to the harm associated with cesarean sections [33].

Regarding the study's limitations, the secondary data source has some weaknesses, such as incomplete data and data completion quality. Nevertheless, increased coverage and progress in completing data related to SIM and SINASC have been pointed out in the literature [34]. Moreover, the present research strengths are that it addresses a cohort of almost two million live births and analyzes factors associated with NNM from a hierarchical approach for entering variables in the multiple model.

The present study identified an association between NNM and sociodemographic, obstetric, childbirth, and healthcare-related factors, reinforcing social disparities and the need to strengthen maternal and child healthcare actions for different population groups. These findings reinforce NNM as a sensitive and strategic public health indicator, as it highlights the interrelationship between social iniquities, weaknesses in perinatal care, and severe neonatal outcomes. Integrating NNM into health planning processes can surveillance capabilities and contribute to more effective public health management. This indicator facilitates the identification of vulnerable territories and populations and the, guiding improvement of maternal and child care networks. Thus, its use can guide intersectoral actions and promote more equitable, evidence-based decision-making. In this context, the study contributes to advancing public policies targeting maternal and child health, particularly in socially vulnerable contexts such as the state of Bahia, and reinforce the commitment to reducing health inequities and improving neonatal outcomes, in alignment with global sustainable development goals.

The findings of this study highlight the strategic value of NNM as a sensitive indicator for health surveillance, public health management, and maternal and child health policy planning. The association between NNM and social determinants underscores the need for targeted interventions to promote equity in perinatal care. Systematic monitoring of NNM can enhance the identification of high-risk populations and geographic areas, support intersectoral actions, and guide the more effective and equitable allocation of healthcare resources. Investing in the quality of prenatal care, childbirth assistance, and the organization of healthcare networks—particularly in socially vulnerable contexts such as the state of Bahia— is essential to reducing health inequities and improve neonatal outcomes, in alignment with the commitments of the Sustainable Development Goals.

## Author contributions

**Conceptualization:** Daiane Porto Nery, Amanda Cristina de Souza Andrade, Vanessa Moraes Bezerra.

**Data curation:** Daiane Porto Nery, Amanda Cristina de Souza Andrade, Vanessa Moraes Bezerra.

**Formal analysis:** Daiane Porto Nery, Amanda Cristina de Souza Andrade, Vanessa Moraes Bezerra.

**Investigation:** Daiane Porto Nery, Amanda Cristina de Souza Andrade, Daniela Silva Rocha, Vanessa Moraes Bezerra.

**Methodology:** Daiane Porto Nery, Amanda Cristina de Souza Andrade, Daniela Silva Rocha, Vanessa Moraes Bezerra.

**Project administration:** Daniela Silva Rocha, Vanessa Moraes Bezerra.

**Software:** Daniela Silva Rocha.

**Supervision:** Amanda Cristina de Souza Andrade, Daniela Silva Rocha, Vanessa Moraes Bezerra.

**Validation:** Daiane Porto Nery.

**Visualization:** Daiane Porto Nery.

**Writing – original draft:** Daiane Porto Nery, Vanessa Moraes Bezerra.

**Writing – review & editing:** Daiane Porto Nery, Amanda Cristina de Souza Andrade, Daniela Silva Rocha, Vanessa Moraes Bezerra.

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
