## [Decision Letter · Decision Letter 0]

2 May 2025

PGPH-D-25-00179

Neonatal Near Miss in a State of Northeastern Brazil: Near Miss Characterization and Determinants

Dear Dr. Bezerra,

Thank you for submitting your manuscript to PLOS Global Public Health. After careful consideration, we feel that it has merit but does not fully meet PLOS Global Public Health’s publication criteria as it currently stands. Therefore, we invite you to submit a revised version of the manuscript that addresses the points raised during the review process.

Your manuscript has been reviewed by two reviewers, whose comments are included below. Please carefully address all comments in your resubmission.

We look forward to receiving your revised manuscript.

Kind regards,

Jennifer Tucker, PhD

Staff Editor

Journal Requirements:

1. We have amended your Competing Interest statement to comply with journal style. We kindly ask that you double check the statement and let us know if anything is incorrect. 2. In the online submission form, you indicated that The data is not in the public domain. It was provided by the Secretary of State of Bahia. Therefore, the data will be made available upon reasonable request. All PLOS journals now require all data underlying the findings described in their manuscript to be freely available to other researchers, either 1. In a public repository, 2. Within the manuscript itself, or 3. Uploaded as supplementary information. This policy applies to all data except where public deposition would breach compliance with the protocol approved by your research ethics board. If your data cannot be made publicly available for ethical or legal reasons (e.g., public availability would compromise patient privacy), please explain your reasons by return email and your exemption request will be escalated to the editor for approval. Your exemption request will be handled independently and will not hold up the peer review process, but will need to be resolved should your manuscript be accepted for publication. One of the Editorial team will then be in touch if there are any issues. 3. We noticed that you used “data not shown”/"unpublished data" in the manuscript. We do not allow these references, as the PLOS data access policy requires that all data be either published with the manuscript or made available in a publicly accessible database. Please amend the supplementary material to include the referenced data or remove the references.

Additional Editor Comments (if provided):

Reviewers' comments:

Reviewer's Responses to Questions

**Comments to the Author**

1. Does this manuscript meet PLOS Global Public Health’s publication criteria ? Is the manuscript technically sound, and do the data support the conclusions? The manuscript must describe methodologically and ethically rigorous research with conclusions that are appropriately drawn based on the data presented.

Reviewer #1: Yes

Reviewer #2: Yes

2. Has the statistical analysis been performed appropriately and rigorously?

Reviewer #1: Yes

Reviewer #2: Yes

3. Have the authors made all data underlying the findings in their manuscript fully available (please refer to the Data Availability Statement at the start of the manuscript PDF file)?

Reviewer #1: No

Reviewer #2: Yes

4. Is the manuscript presented in an intelligible fashion and written in standard English?

Reviewer #1: Yes

Reviewer #2: No

5. Review Comments to the Author

Reviewer #1: Dear authors, I would like to congratulate you on the completion of your article.

The manuscript meets the PLOS Global Public Health criteria for publication. The research is methodologically rigorous, the data support the conclusions, and the study is ethically sound. Statistical analysis was performed appropriately and rigorously. The authors used appropriate statistical tests, and the results are presented clearly and concisely. The manuscript is presented in an understandable manner and written in standard English. However, there are some grammatical and typographical errors that need to be corrected. The study is well written and the results are interesting. However, the authors need to address the following issues:

If possible, make all data underlying the findings available in your manuscript.

Correct any grammatical and typographical errors.

Discuss the public health implications of the results.

Reviewer #2: Dear Author.

Following are my comments

1. In you manuscript methodology, Neonatal near miss has been defined as any one of the followings: Birth weight < 1500 gm, Gestational age < 32 wks, Apgar score at 5 mins < 7 and Congenital malformation.

Could you explain what are the babies number invidiously.

2. What is the mean birth weight, mean gestational age and mean 5th minute Apgar score.

6. PLOS authors have the option to publish the peer review history of their article (what does this mean? ). If published, this will include your full peer review and any attached files.

**Do you want your identity to be public for this peer review?** For information about this choice, including consent withdrawal, please see our Privacy Policy .

Reviewer #1: **Yes:** Leandro Alves da Luz

Reviewer #2: **Yes:** Prof. Dr Sunil Raja Manandhar

---

## [Decision Letter · Decision Letter 1]

19 Jun 2025

PGPH-D-25-00179R1

Near Miss Neonatal em um Estado do Nordeste Brasileiro: Caracterização e Determinantes do Near Miss

Dear Dr. Andrade,

Thank you for submitting your manuscript to PLOS Global Public Health. After careful consideration, we feel that it has merit but does not fully meet PLOS Global Public Health’s publication criteria as it currently stands. Therefore, we invite you to submit a revised version of the manuscript that addresses the points raised during the review process.

Please see the reviewers' comments below. I believe reviewer 1's comment about fine terminology consistency refers to the use of both "NMN" (near miss neonatal) and "NNM" (neonatal near miss) throughout your manuscript. Please carefully address the reviewer comments and provide a point-by-point response to reviewers upon resubmission.

We look forward to receiving your revised manuscript.

Kind regards,

Sarah Jose, Ph.D.

Staff Editor

Journal Requirements:

Additional Editor Comments (if provided):

Reviewers' comments:

Reviewer's Responses to Questions

**Comments to the Author**

1. If the authors have adequately addressed your comments raised in a previous round of review and you feel that this manuscript is now acceptable for publication, you may indicate that here to bypass the “Comments to the Author” section, enter your conflict of interest statement in the “Confidential to Editor” section, and submit your "Accept" recommendation.

Reviewer #1: All comments have been addressed

Reviewer #2: All comments have been addressed

2. Does this manuscript meet PLOS Global Public Health’s publication criteria ? Is the manuscript technically sound, and do the data support the conclusions? The manuscript must describe methodologically and ethically rigorous research with conclusions that are appropriately drawn based on the data presented.

Reviewer #1: Yes

Reviewer #2: Yes

3. Has the statistical analysis been performed appropriately and rigorously?

Reviewer #1: Yes

Reviewer #2: Yes

4. Have the authors made all data underlying the findings in their manuscript fully available (please refer to the Data Availability Statement at the start of the manuscript PDF file)?

Reviewer #1: No

Reviewer #2: Yes

5. Is the manuscript presented in an intelligible fashion and written in standard English?

Reviewer #1: Yes

Reviewer #2: Yes

6. Review Comments to the Author

Reviewer #1: The manuscript “Neonatal Near Miss in a State of Northeastern Brazil: Characterization and Determinants of Near Miss” (PGPH-D-25-00179R1) showed substantial progress in response to the previous round of review. The authors conducted a thorough review, addressing grammatical, typographical, and stylistic issues, which resulted in a notable improvement in the clarity and coherence of the text. The language was refined to meet scientific writing standards, and there is greater terminological consistency. The methodology is sound, and the statistical analyses appear rigorous and appropriate for the data presented. The findings are relevant to global public health, especially in the Brazilian context.

The remaining changes that warranted a “Minor Revision” are specific and focus on improvements that do not require significant restructuring of the study or additional complex analyses. Points to be addressed include:

Fine terminological consistency: A final passage to ensure absolute consistency of key terms (such as “near miss neonatal” vs. “near miss neonatal”) throughout the manuscript.

Highlighting public health implications: Although the discussion already addresses the implications, a more concise final paragraph focused on actionable recommendations derived from the results would strengthen the impact of the Discussion section.

Overall, the manuscript is in excellent shape, and the above suggestions are intended only to refine specific aspects to maximize its clarity and full adherence to publication requirements.

Reviewer #2: Dear Author,

Thank you for submitting your revision . Now your manuscript looks ok.

7. PLOS authors have the option to publish the peer review history of their article (what does this mean? ). If published, this will include your full peer review and any attached files.

**Do you want your identity to be public for this peer review?** For information about this choice, including consent withdrawal, please see our Privacy Policy .

Reviewer #1: **Yes:** Leandro Alves da Luz

Reviewer #2: **Yes:** Prof Dr SUnil Raja Manandhar

---

## [Decision Letter · Decision Letter 2]

30 Jul 2025

Neonatal Near Miss in a State of Northeastern Brazil: Near Miss Characterization and Determinants

PGPH-D-25-00179R2

Dear Dr. Bezerra,

We are pleased to inform you that your manuscript 'Neonatal Near Miss in a State of Northeastern Brazil: Near Miss Characterization and Determinants' has been provisionally accepted for publication in PLOS Global Public Health.

Best regards,

Julia Robinson

Executive Editor

Reviewer Comments (if any, and for reference):

Reviewer's Responses to Questions

**Comments to the Author**

1. If the authors have adequately addressed your comments raised in a previous round of review and you feel that this manuscript is now acceptable for publication, you may indicate that here to bypass the “Comments to the Author” section, enter your conflict of interest statement in the “Confidential to Editor” section, and submit your "Accept" recommendation.

Reviewer #1: All comments have been addressed

Reviewer #2: All comments have been addressed

2. Does this manuscript meet PLOS Global Public Health’s publication criteria ? Is the manuscript technically sound, and do the data support the conclusions? The manuscript must describe methodologically and ethically rigorous research with conclusions that are appropriately drawn based on the data presented.

Reviewer #1: Yes

Reviewer #2: Yes

3. Has the statistical analysis been performed appropriately and rigorously?

Reviewer #1: Yes

Reviewer #2: Yes

4. Have the authors made all data underlying the findings in their manuscript fully available (please refer to the Data Availability Statement at the start of the manuscript PDF file)?

Reviewer #1: Yes

Reviewer #2: No

5. Is the manuscript presented in an intelligible fashion and written in standard English?

Reviewer #1: Yes

Reviewer #2: Yes

6. Review Comments to the Author

Reviewer #1: I would like to commend the authors for the improvements made in this revised version of the manuscript. The study addresses a relevant and timely topic—the characterization and determinants of neonatal near miss (NNM)—with methodological rigor and comprehensive analysis based on a large and robust dataset from Bahia, Brazil. The use of hierarchical modeling, a clear conceptual framework, and consistent interpretation of the findings contributes significantly to the field of maternal and child health, particularly in contexts of social vulnerability.

The manuscript is technically sound, the conclusions are well supported by the data, and the implications for public health policy are well articulated. The authors also appropriately addressed the data availability policy and ethical concerns.

Reviewer #2: Dear Author,

My issues has been addressed. Now it ok.

7. PLOS authors have the option to publish the peer review history of their article (what does this mean? ). If published, this will include your full peer review and any attached files.

**Do you want your identity to be public for this peer review?** For information about this choice, including consent withdrawal, please see our Privacy Policy .

Reviewer #1: **Yes:** LEANDRO ALVES DA LUZ

Reviewer #2: **Yes:** Prof Dr Sunil Raja Manandhar
